# Effect of Combined Exercise Program on Lower Extremity Alignment and Knee Pain in Patients with Genu Varum

**DOI:** 10.3390/healthcare11010122

**Published:** 2022-12-30

**Authors:** Hyung-Hoon Moon, Yong-Gon Seo, Won-Moon Kim, Jae-Ho Yu, Hae-Lim Lee, Yun-Jin Park

**Affiliations:** 1Department of Sports Medicine, Cha University, Gyeonggi-do, Pocheon-Si 11160, Republic of Korea; 2Samsung Medical Center, Department of Orthopedic Surgery, Division of Sports Medicine, Seoul 06351, Republic of Korea; 3Department of Sports Science, Dongguk University, Gyeongju-si 38066, Republic of Korea; 4Department of Physical Therapy, Sunmoon University, Asan 31460, Republic of Korea; 5Division of Health Rehabilitation, Osan University, 45, Cheonghak-ro, Osan-si 18119, Republic of Korea

**Keywords:** genu varum, middle-aged women, lower extremity, knee to knee length, hip–knee–ankle angle, hip inclination angle, knee pain, combined exercise

## Abstract

This study aimed to assess the effect of a combined exercise program on lower-extremity alignment and knee pain in patients with genu varum. Forty-seven middle-aged women with knee pain and genu varum were randomly divided into the exercise (EG, *n* = 24) and control (CG, *n* = 23) groups. The exercise group underwent a combined exercise program lasting 60 min in one session, three times a week for 12 weeks. Knee-to-knee length (KTKL), hip-knee-ankle angle (HKAA), hip inclination angle (HIA), and medial proximal tibial angle (MPTA) were assessed to evaluate lower-extremity alignment. To evaluate knee pain, the short form-McGill Pain Questionnaire (SF-MPQ) were used. There was a significant difference between the groups, and a decrease of 16% in KTKL (from 6.48 ± 1.26 cm to 5.47 ± 1.21 cm) was shown in EG. Other variables, including HKAA, HIA, and MPTA on the right side, showed significant differences between pre- and post-intervention in EG (*p* < 0.01, *p* < 0.01, and *p* < 0.01, respectively). SF-MPQ score improved with 45% from 18.75 ± 1.64 to 10.33 ± 2.47 after exercise intervention in EG. These results suggest that the combined exercise program, including strength and neuromuscular exercises, is an effective intervention for improving lower-extremity alignment and knee pain in middle-aged women with genu varum.

## 1. Introduction

Knee-joint stability is determined by the movement path of the lower-extremity axis and the slope of the knee-joint surface [1] and is also achieved by the soft tissue around the patella. This stability affects the hip joint above it and the ankle joint below it [2]. This means that the structural instability of the knee joint by affecting the hip and ankle joints leads to lower-extremity malalignment, which causes structural deformation of the knee joint, damage to the surrounding ligaments or cartilage, and knee osteoarthritis [3,4]. Genu varum is a common lower-extremity malalignment. In middle-aged postmenopausal women, particularly those characterized by muscle weakness and reduced bone density, genu varum progresses rapidly, which may lead to pain, loss of function, and reduced quality of life [5].

In patients with genu varum, tension in the medial rotator muscle increases due to medial deviation of the patella and medial rotation of the femur, which leads to decreased momentum and weakness of the lateral rotator muscle [6]. These structural deformations and muscle imbalance cause external rotational deformation of the tibia and talipes varus. To compensate for this, the angle between the femur and tibia increases, further widening the length between the legs [7,8].

The knee joint is the most common weight-bearing joint, and osteoarthritis (OA) predominantly occurs in the medial compartment of the tibiofemoral joint [9]. Patients with OA frequently have varus knees and complain commonly of knee pain and stiffness, as well as difficulties with activities of daily living [10,11]. OA in the knee joint is associated with a change in biomechanical aspects, including an increase in activity such as knee adduction moments during walking [12]. Therefore, an improvement in lower-extremity alignment through treatment interventions is needed to improve structural problems and muscle imbalance.

Exercise therapy is an option for improving knee function and alignment of the lower extremities [13,14]. An appropriate exercise program is a valuable strategy for reducing pain due to musculoskeletal disorders, and increasing physical activity could eliminate an important risk factor, sedentary lifestyle, which promotes worsening pain [15]. A previous study [14] reporting on corrections for genu varum demonstrated that lower-extremity strength exercises were an effective intervention to reduce knee pain and to improve physical function. Neuromuscular training has a very important role in improving alignment between the trunk and lower extremities [13]. However, only a few authors have studied the effect of strength exercise on structural improvement and on improvements in physical function and muscle strength by applying neuromuscular exercises [13,14].

Most previous studies [13,14,15,16] reporting on improvements in alignment have been conducted by applying single-exercise programs to confirm the effects of exercise interventions on pain or function in genu varum. A previous study [17] demonstrated that a combined exercise program is a feasible and effective intervention for improving muscle activation, movement patterns, and alignment.

However, there are currently no studies on the effect of a combined exercise program for improving structural change, confirmed by improvements in alignment and knee pain in patients with genu varum but not OA. Therefore, this study aimed to investigate the effect of a combined exercise program including strength and neuromuscular exercises on lower-extremity alignment and knee pain in patients with genu varum.

## 2. Materials and Methods

### 2.1. Participants and Study Design

A total of 47 female patients (age 50–60 years) with genu varum and a short form-McGill Pain Questionnaire (SF-MPQ) score of 18.0 or higher were included from the P Department of Rehabilitation Medicine in Seoul, Korea. According to the orthopedic surgeon’s diagnosis, their Kellgren–Lawrence Grades were from 0 to 1, and the study was limited to participants who were able to walk normally despite pain but was suspected to have a joint space that was narrower than normal and had minor cartilage damage [18].

The participants were randomly divided into an exercise group (EG, *n* = 24) and a control group (CG, *n* = 23). Those who had neurological damage within 3 months, such as surgical history, radiant pain, and blunt sensation, or who had undergone exercise therapy for genu varum were excluded (Figure 1). The general characteristics of the participants are presented in Table 1.

### 2.2. Radiographic Evaluation

For whole-leg X-ray examination of the lower extremities (INNOVISION ver. 2.0 program, DK Medical Systems Co., Pyeongtaek, Korea), the participants were asked to stand in an upright position while watching the front, while their inside malleoluses of the tibia from both ankles contacted each other. Knee-to-knee length (KTKL) was measured as the distance between the medial epicondyle of the knee joints in the upright position and recorded in units of 0.1 cm [19]. Hip–knee–ankle angle (HKAA) was measured as the medial angle between the lines from the center of the femoral head to the midpoint between the tips of the tibial spines and the superior facet of the talus (Figure 2A). An HKAA of less than 180° was diagnosed as genu varum [20]. Hip inclination angle (HIA) was measured as the angle between two lines: a line connecting 10 cm below the top of the femur neck and 10 cm above the lower center of the femur and a line connecting the thinnest point of the femoral neck and the tip of the femoral head (Figure 2B). HIA less than 125°, a normal angle, was diagnosed as genu varum [21]. Medial proximal tibial angle (MPTA) was measured as the angle between the tibial axis and knee joint line of the proximal tibia (Figure 2C). An MPTA less than 90° was diagnosed as genu varum [22].

### 2.3. SF-MPQ

The SF-MPQ is a scaled questionnaire consisting of 22 items rated on a scale of 0 to 10 and has 3 subscales: an 11-item sensory subscale, a 4-item affective subscale, and a 7-item pain intensity subscale [23]. The sum of the scores on the three subscales was used for the analysis.

### 2.4. Combined Exercise Program

The exercise program was conducted 3 times a week for 12 weeks, and each session lasted 60 min in the following order: warm-up, main exercise, and cool-down. Exercise intensity was measured using the Borg Rating of Perceived Exertion (RPE) 6–20 scale, set to RPE 11–13 for warm-up and cool-down, and RPE 14–16 for the main exercise, and was increased gradually. Resting time was set to 60 s between events and 30 s between sets.

The CG was treated with therapy consisting of thermal stimulation using an electric heat pack and ultrasound therapy. This thermal therapy was applied using an electric heat pack for 25 min by setting the temperature at 50° to 55°, and ultrasound therapy used an Inter current therapy (ITO, Seoul, Korea) for 15 min at 100 Hz, a frequency of 1 MHz, and an intensity of 1.5 W/cm^2^ for 10 min, with a total treatment period of 50 min. Table 2 shows the details of the program.

### 2.5. Data Analysis

All variables are expressed as mean and standard deviation using descriptive statistics. A two-way repeated ANOVA was used to analyze the interaction and difference between the two groups. The paired *t*-test was used for a comparison of the results between pre- and post-test in each group. For the comparison between the group, an independent *t*-test was performed. A data analysis was performed using Windows SPSS/PC version 21.0 (IBM Inc., Chicago, IL, USA). *p*-value < 0.05 was considered significant difference.

## 3. Results

In changes in KTKL, the results showed significant difference between the groups (*F* = 9.872, *p* < 0.01), time (*F* = 17.972, *p* < 0.001) and interaction (*F* = 34.554, *p* < 0.001) of KTKL (Table 3). In the result of independent t-test, there was a significant difference between the group (*p* < 0.001) (Figure 3).

In the results of HKAA, the left side showed significant differences in time (*F* = 11.635, *p* < 0.01) and interaction (*F* = 34.542, *p* < 0.001) between groups but no significant difference between the group (*F* = 2.462 *p* = 0.124). The right side in HKAA showed significant differences between the groups (*F* = 4.831, *p* < 0.05), time (*F* = 15.775, *p* < 0.001), and interaction (*F* = 29.524, *p* < 0.001) (Table 3). In the results of the independent *t*-test, HKAA on both sides showed significant differences between the group (*p* < 0.01) (Figure 3).

In the changes in HIA, the left side showed significant differences in time (*F* = 17.908, *p* < 0.001) and interaction (*F* = 37.089, *p* < 0.001) between the groups but no significant differences between the group (*F* = 0.042, *p* = 0.838). The right side showed a significant difference in time (*F* = 12.368, *p* < 0.01) and interaction (*F* = 23.058, *p* < 0.001) but no significant differences between the group (*F* = 0.191, *p* = 0.664) (Table 3). There was no a significant difference between the group in HIA of the right or left sides (*p* = 0.178, *p* < 0.259, respectively), and improvement was shown only after intervention in EG (*p* < 0.001) (Figure 3).

In the results of MPTA, the left side showed significant differences between the groups (*F* = 21.039, *p* < 0.001), time (*F* = 5.993, *p* < 0.05), and interaction (*F* = 27.076, *p* < 0.001). The right side showed significant differences in time (*F* = 5.224, *p* < 0.05) and interaction (*F* = 52.953, *p* < 0.001), but no a significant difference between the group (*F* = 0.001, *p* = 0.980) (Table 3). In the results of the independent *t*-test, MPTA in the left side showed a significant difference between the group (*p* < 0.01) that was not shown in right side (*p* = 0.475) (Figure 3).

In the change in SF-MPQ, when measuring knee pain, significant differences were shown in the groups (*F* = 29.160, *p* < 0.001), time (*F* = 95.671, *p* < 0.001), and interaction (*F* = 68.686, *p* < 0.001) (Table 3). There was a significant difference between the group according to the analysis of the independent *t*-test (*p* < 0.001) (Figure 3).

## 4. Discussion

This study was conducted to confirm the effects of a combined exercise program on lower-extremity alignment in genu varum and demonstrated that exercise is a beneficial intervention to improve alignment in the lower extremities, confirmed by KTKL, HKAA, HIA, and MPTA.

Genu varum, malalignment of the lower extremities, may cause an instability of the lower extremities, resulting in spinal deformity, joint degeneration, and gait disturbance. Patellar position is associated with knee disease, and medial deviation of the patella may lead to knee pain and osteoarthritis [24].

Various variables, including KTKL, HKAA, HIA, and MPTA, are used for evaluating knee joint and lower-extremity alignment. KTKL is the interval between the medial part of the knee joint, and the variable was used in this study to evaluate the severity of genu varum. In the present study, KTKL showed a significant decrease of 16% from 6.48 ± 1.26 cm to 5.47 ± 1.21 cm. This result is consistent with the results of Kwon et al. [25], who applied elastic band exercises and stretching as their treatment of choice, and another study used resistance exercise and stretching [26]. Unlike previous studies [25,26], this study showed that combined exercises including neuromuscular exercises is effective for improving KTKL.

HKAA is to measure lower-limb alignment and measured from a full-length lower extremity radiograph. In our study, HKAA on right and left sides improved after combined exercise intervention in the EG. A previous study conducted by Park et al. [27] showed that 3 months of hip-external-rotator strength training significantly improved HKAA in patients with genu varum. We hypothesized that the combined exercise in this study would affect lower-extremity alignment through improvements in muscle flexibility, range of motion, and hip-joint muscle strength. However, further study is needed to compare the difference in HKAA between the used exercise types. HIA is usually 125 ± 5° in a normal adult, and the mean value in this study was lower than the normal reference. The result of the present study demonstrated that this combined exercise resulted in a significant improvement of HIA in EG but not CG and that there was no difference between the group. There is no current study on the effect of exercise intervention on the change in HIA, but the finding in this study may be considered for the further study.

MPTA is one of the most common variables used to measure the severity of OA and to can be simply measured from knee radiographs. Most previous studies [28,29] have reported that MPTA is improved through surgery such as total knee arthroplasty and high tibial osteotomy. A study has reported that corrective exercises including strength and mobilization exercises will help improve MPTA and functional tasks in knee joint. In our study, the combined exercise program showed improvements in MPTA after 3 months of intervention in EG. We conclude that it is meaningful to study the change in MPTA by applying exercise therapy.

This study analyzed changes in pain after exercise in patients with genu varum. The results showed significant differences between the group in pain, which is consistent with the results of Bennell et al. [30], who reported that neuromuscular and quadriceps strengthening exercises reduced knee pain. Several studies [31,32] demonstrated that exercise intervention such as quadriceps strengthening is effective at reducing knee pain and at improving function in patients with knee pain.

However, the group with varus malalignment did not appear to experience effective reductions in pain [31,32]. Recent international evidence-based guidelines for knee OA recommend a multimodal treatment approach with patient education as one of the cornerstones of treatment [33]. The combined exercise in this study consisted of various exercises, focusing on correcting muscle imbalance caused by genu varum. We believe that the proposed exercises improved the static stability and strength of the pelvis and lower extremities; alleviated muscle imbalances, possibly by strengthening the large lower extremity muscle groups (quadriceps femoris and rectus femoris muscles) through resistance exercise; and significantly decreased the knee pain caused by body movements by balancing the dynamic load within the knee joint.

## 5. Limitation

This study has some limitations. First, the study had a small sample size although it was divided into two different groups. Thus, it is difficult to apply these results to all patients with genu varum. A large sample study is needed to apply these results. Secondly, this study did not measure the hip rotator strength despite hip-rotator strengthening exercises being applied in exercise groups. Therefore, further study is required to confirm the effect of hip-rotator strength on these results.

## 6. Conclusions

This study evaluated the effect of a complex exercise program on reducing lower extremity malalignment and pain and on improving function in patients with genu varum accompanied by pain. The results of this study support the idea that a combination of various exercises is more effective than current exercise programs in improving genu varum.

## Figures and Tables

**Figure 1 healthcare-11-00122-f001:**
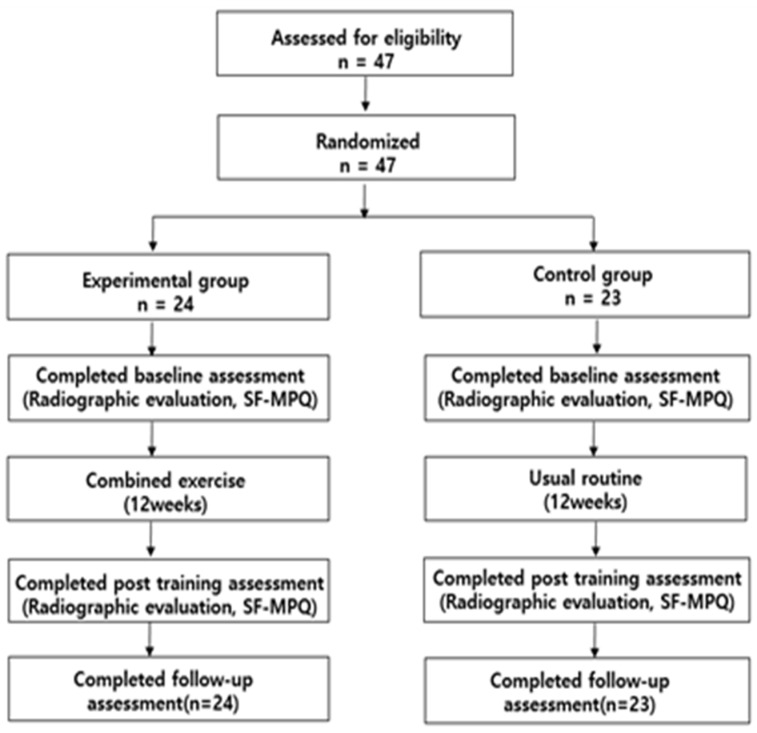
A flowchart of this study.

**Figure 2 healthcare-11-00122-f002:**
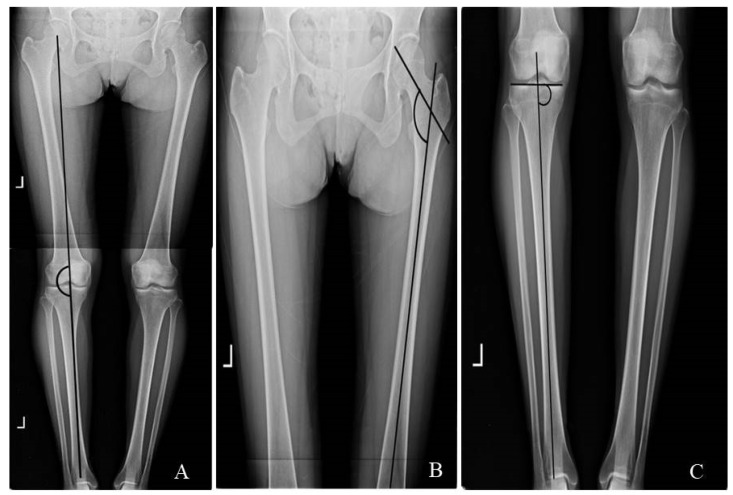
Measurement of lower extremity alignment: (**A**) measurement of hip–knee–ankle angle; (**B**) measurement of hip inclination angle; (**C**) measurement of medial proximal tibia angle.

**Figure 3 healthcare-11-00122-f003:**
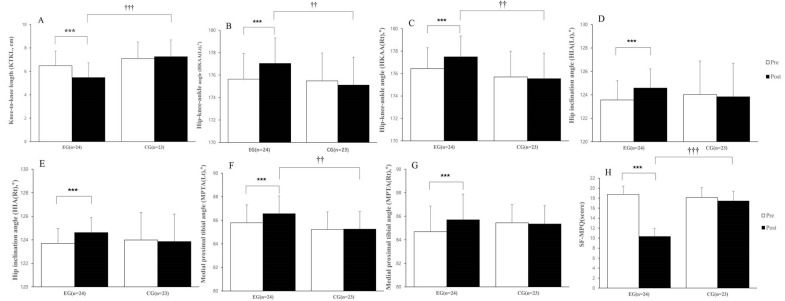
Values are presented as the mean ± standard deviation: (**A**) knee-to-knee length (KTKL); (**B**) hip–knee–ankle angle (**Left**); (**C**) hip–knee–ankle angle (**Right**); (**D**) hip inclination angle (**Left**); (**E**) hip inclination angle (**Right**); (**F**) medial proximal tibial angle (**Left**); (**G**) medial proximal tibial angle (**Right**); (**H**) short form-McGill Pain Questionnaire (SF-MPQ). Significant differences using paired *t*-test: *** *p* < 0.001. Significant differences using independent *t*-test: †† *p* < 0.01, ††† *p* < 0.001.

**Table 1 healthcare-11-00122-t001:** Physical characteristics of participants.

Variables	Exercise Group (=24)	Control Group (=23)	*p*-Value
Age (years)	53.15 ± 2.59	53.04 ± 2.82	0.68
Height (cm)	161.92 ± 3.55	163.21 ± 4.26	0.15
Weight (kg)	53.53 ± 5.09	56.57 ± 7.24	0.29
BMI (kg/m^2^)	20.25 ± 1.92	21.29 ± 2.21	0.96
SF-MPQ	18.75 ± 1.64	18.13 ± 1.98	0.59

Values are mean ± standard deviation. BMI: body mass index, SF-MPQ: short form-McGill Pain Questionnaire.

**Table 2 healthcare-11-00122-t002:** Combined exercise program.

Division	Workout Types	Intensity	Time	Set
Warm-up	Treadmill: gait corrective program	RPE 11–13	5 min	
60 min combined exercise for 12 weeks: 3 days/week	Release of major joint and muscle group - Band stretch : hamstring, gastrocnemius, hip adductor - Form roller : tibialis anterior, anterior capsular ligament, tensor fascia latae, iliopsoas	RPE 11–13	10 min	Each exercise 3 set/1 set 10 R (10 s stop)
	Resistance exercise - Leg extension: quadriceps (VMO) - Smith machine lunge and wall squat - Wall bar squat - Hip external rotator	RPE 14–16	15 min	Each exercise 3 set/1 set 15–20 R
	Neuromuscular exercise - Forward and backward sliding or stepping - Sideways exercises - Functional hip muscle strengthening - Functional knee muscle strengthening - Step-ups and down - Balance board exercise		15 min	Each exercise 3 set/1 set 20–30 R
	Stylex Q - Hip external rotator		10 min	1 set/100 R
Cool-down	Treadmill: gait corrective program	RPE 11–13	5 min	

RPE: rating of perceived exertion, VMO: vastus medialis oblique.

**Table 3 healthcare-11-00122-t003:** Changes in structural variables and pain in patients with genu varum.

Variables	Groups	Pre-Intervention	Post-Intervention		*F*-Value	*p*-Value	η2
KTKL (cm)	EG	6.48 ± 1.26	5.47 ± 1.21	G T GxT	9.872 17.192 34.554	0.003 ** 0.000 *** 0.000 ***	0.180 0.276 0.434
	CG	7.08 ± 1.43	7.25 ± 1.44				
HKAA(Lt) (degree)	EG	175.64 ± 2.27	177.04 ± 2.11	G T GxT	2.462 11.635 34.542	0.124 0.001 ** 0.000 ***	0.052 0.205 0.434
	CG	175.48 ± 2.49	175.11 ± 2.41				
HKAA(Rt) (degree)	EG	176.44 ± 1.86	177.49 ± 2.01	G T GxT	4.831 15.775 29.524	0.033 * 0.000 *** 0.000 ***	0.097 0.260 0.396
	CG	175.70 ± 2.28	175.54 ± 2.30				
HIA(Lt) (degree)	EG	123.56 ± 1.64	124.58 ± 1.37	G T GxT	0.042 17.908 37.089	0.838 0.000 *** 0.000 ***	0.001 0.285 0.452
	CG	124.02 ± 2.86	123.84 ± 2.82				
HIA(Rt) (degree)	EG	123.69 ± 1.27	124.62 ± 1.10	G T GxT	0.191 12.368 23.058	0.664 0.001 ** 0.000 ***	0.004 0.216 0.339
	CG	123.99 ± 2.33	123.85 ± 2.51				
MPTA(Lt) (degree)	EG	85.79 ± 1.51	86.56 ± 1.18	G T GxT	21.039 5.993 27.076	0.000 *** 0.018 * 0.000 ***	0.319 0.118 0.376
	CG	85.22 ± 1.49	85.25 ± 1.70				
MPTA(Rt) (degree)	EG	84.69 ± 2.18	85.70 ± 1.98	G T GxT	0.001 5.224 52.953	0.980 0.027 * 0.000 ***	0.000 0.104 0.541
	CG	85.44 ± 1.55	85.35 ± 1.42				
SF-MPQ (score)	EG	18.75 ± 1.64	10.33 ± 2.47	G T GxT	29.160 95.671 68.686	0.000 *** 0.000 *** 0.000 ***	0.393 0.680 0.604
	CG	18.13 ± 1.98	17.43 ± 3.81				

Values are presented as the mean ± standard deviation. EG, exercise group; CG, control group; KTKL, knee-to-knee length; HKAA, hip–knee–ankle angle; HIA, hip inclination angle; MPTA, medial proximal tibial angle; SF-MPQ, short form-McGill Pain Questionnaire; Rt, right; Lt, left. The results for the two-way repeated measure ANOVA. * *p* < 0.05, ** *p* < 0.01, *** *p* < 0.001.

## Data Availability

The data used to support the findings of this study are available from the corresponding author upon reasonable request.

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
