# Peer review of "Effect of Combined Exercise Program on Lower Extremity Alignment and Knee Pain in Patients with Genu Varum"

_healthcare, 2022, doi:10.3390/healthcare11010122_

Round 1

Reviewer 1 Report

Dear Author, I would like to thank for your efforts. When I evaluated your research in detail, I have identified some major corcens.

Introduction

You need to include a reference in Lines 53-55. (which previous studies ?)

Methods

Please add study design.

You need to apply repeated measures ANOVA if you have a pre-posttest in two groups. The error rate of the t-test is higher than acceptable, so I recommend you reanalyze the data. Also, please add effect size.

You could better put the figures (1, 2, 3) side by side.

Result

What was the point of these parameters? Why were strength parameters not included? Explain please.

Table 3 can be also presented graphically for more readability.

Discussion

The discussion section is shorter than a well-structured one and constructed with a few references, so it would be better to expand it with up-to-date literature.

Please provide a new section for limitations rather than discussion.

Reviewer 2 Report

The authors describe that a combined exercise program could improve lower extremity alignment and knee pain in middle-aged women with genu varum. The whole design is clear and straightforward. And I have some concerns as following:

1.The introduction could be expanded.

2.In table 3, there are mixed bold font, "Variable ", "KTKL", and underline.

3.Authors could explain more about "between-group", it means "Post-intervention" in Exercise group vs. "Post-intervention" in Control group, right?

4. Authors could discus more why MPTA(Lt) showed significant between-group difference, while MPTA(Rt) showed no significant between-group difference in the EG.

Reviewer 3 Report

The article Effect of combined exercise program on lower extremity alignment and knee pain in patients with genu varum is well written and well organized. Limb malalignment is an important topic and therapeutic approach is yet to be established. However, the introduction suggest that genu varum is a cause of osteoarthritis, whereas in most cases it is an effect of arthritic changes in medial compartment of the knee joint and articular cartilage degeneration. Connection between knee malalignment and OA should be more clearly presented in the introduction. The introduction is coherent and precise however I feel that some elaboration on OA, which is most common cause of genu varum in adults should be introduced. Authors may find some useful information in the works: DOI 10.1016/S0140-6736(19)30417-9; https://doi.org/10.1136/annrheumdis-2013-204763; DOI 10.3390/app11041552; DOI 10.3390/app9194102;

The methodology adopted in the paper is described succinctly and clearly.  The results have been discussed in a  correct manner allowing for easy interpretation.

I have not found radiological evaluation of OA in the study group, what was the KL grade?

In discussion other treatment modalities for OA and genu varum should be mentioned such as for instance HTO or joint preserving procedures, should those procedures be combined with the designed exercise program? DOI 10.1155/2019/8363128; 10.3390/app10238312;

The conclusions are supported by the results obtained were presented correctly in a concise manner.

After making the appropriate additions, the article may be accepted for publication. The results obtained can find practical application in daily clinical practice.

Round 2

Reviewer 1 Report

Thanks for your effort.

Author Response

Dear reviewer

The authors really thank you for your comments, which was very helpful in improving the quality of our manuscript.

Thanks again.